# Learning the Geometry of Wave-Based Imaging

**Konik Kothari**
UIUC
kkothar3@illinois.edu

**Maarten de Hoop**
Rice University
mvd2@rice.edu

**Ivan Dokmanić**
University of Basel
ivan.dokmanic@unibas.ch

## Abstract

We propose a general physics-based deep learning architecture for wave-based imaging problems. A key difficulty in imaging problems with a varying background wave speed is that the medium "bends" the waves differently depending on their position and direction. This space-bending geometry makes the equivariance to translations of convolutional networks an undesired inductive bias. We build an interpretable neural architecture inspired by Fourier integral operators (FIOs) which approximate the wave physics. FIOs model a wide range of imaging modalities, from seismology and radar to Doppler and ultrasound. We focus on learning the geometry of wave propagation captured by FIOs, which is implicit in the data, via a loss based on optimal transport. The proposed FIONet performs significantly better than the usual baselines on a number of imaging inverse problems, especially in out-of-distribution tests.

## 1   Introduction

We propose a deep learning approach for wave-based imaging with applications ranging from medical photoacoustic tomography to reflection seismology. A simple intuition for imaging with waves can be gleaned from Figure 1. Elementary wave packets propagate from where they are created (sources $S_{1,2,3}$ in 1A), and then possibly scattered (an interface, 1D), to where they are sensed. When and where a wave packet arrives at a sensor (1B) depends on its orientation and position and the geometry associated with the background wave speed. To the first approximation, imaging is accomplished by routing the wave packets back where they were created or scattered.

We consider the problem of estimating an image, $v$ from measurements, $u$ obtained by an imaging forward operator, $A_\sigma$ given as

$$u = A_\sigma v \ (\text{+errors}). \tag{1}$$

The unknown $v$ could, for example, represent the interfaces within the subsurface of Earth. The forward operator (and, hence, its inverse) is parameterized by $\sigma$. Here, $\sigma$ is the background wave speed, a material parameter of the medium. Note that $\sigma$ varies across the domain and controls the ray paths of wave packets (see Figure 1A,D); thus defining a

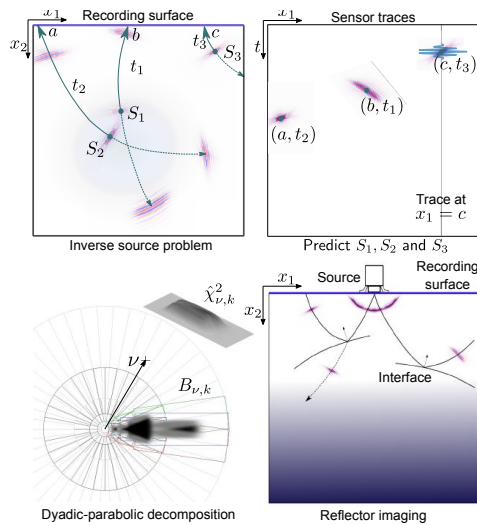

Figure 1: *A:* Sources $S_1, S_2$ and $S_3$ are recorded at the surface at times $t_1$, $t_2$ and $t_3$. *B:* The recording surface generates sensor traces that are used to image the sources. *C:* Wave packets are localized in frequency by directional bandpass filters $\chi^2_{\nu,k}$. *D:* In reflection imaging, reflections of waves are recorded on the surface (see Appendix A.6).

"geometry" that enables wave-based imaging. $A_\sigma$ depends on $\sigma$ in a strongly nonlinear fashion. In this work we assume that $\sigma$, and hence $A_\sigma$, is unknown—we only know $A_\sigma$ up to a class.

We aim to estimate the inverse of (1) by a neural network $f$ trained with a loss $L$. The central question is how to design $f$ and $L$. Our approach is based on the physics of wave propagation as captured by the Fourier integral operators (FIOs) with a loss based on optimal transport. As a consequence, our network exhibits strong out-of-distribution (OoD) generalization and improves interpretability. FIOs describe a vast variety of imaging modalities including reflection seismology [1, 22], thermoacoustic [28, 44] and photoacoustic [17] tomography, radar [3, 35, 10], and single photon emission computed tomography [23], modeling both forward and inverse maps. Therefore, our network design is applicable for many imaging modalities.

An FIO $F_\sigma$ maps the input $u \in L_2(\mathbb{R}^2)$ (for example, a record of pressure time traces (see Figure 1B)) to its output $F_\sigma[u]$ (for example, an image of a human brain), as

$$F_\sigma[u](y) = \frac{1}{(2\pi)^2} \int_{\mathbb{R}^2} a(y, \xi) \hat{u}(\xi) e^{i S_\sigma(y, \xi)} d\xi, \tag{2}$$

where $\hat{u}$ denotes the Fourier transform of $u$, $a(x, \xi)$ is called the symbol of $F_\sigma$, and $S_\sigma(x, \xi)$ is a suitable phase function (cf. Section 2.2). FIOs are a natural extension of convolutions. If $S(x, \xi) = \langle x, \xi \rangle$ and $a(x, \xi) \equiv \hat{a}(\xi)$, (2) is indeed a convolution; it models simple deblurring or denoising. Allowing a general $a(x, \xi)$ makes it a pseudodifferential operator; these appear as approximate solutions of elliptic PDEs [46] or normal operators of imaging [27]. For a general phase $S(x, \xi)$, $F$ becomes powerful: it can deform the domain in an orientation-dependent way. This models approximate solutions of hyperbolic PDEs and therefore wave propagation. The geometry of an FIO (Figure 1), dictated by the medium parameter $\sigma$, is completely captured in its phase $S_\sigma(x, \xi)$.

## 1.1  Our results

Our architecture design, based on discretization of FIOs [8, 25], improves interpretability and enables strong out-of-distribution generalization without any additional transfer- or meta-learning schemes [6, 31, 32]. This is essential to imaging in exploratory sciences and medical applications where failing out-of-distribution can be disastrous [5]. A key ingredient that allows this is a module that learns geometry—the *wave packet (WP) routing network*. This module is interpretable in that its output provides physically meaningful deformation maps of the domain. The WP routing network warps pixel grids and never "looks" at pixel intensities. Hence, once trained, it is data-independent. Another key ingredient to learning this geometry is a training strategy and a loss function based on optimal transport.

## 1.2  Relation to prior work

Existing physics-based approaches either substitute forward models into unrolled networks or apply auto-differentiation to spatiotemporal fields parameterized by neural networks [7, 36, 37]. In either case the forward operator should be known in closed form and should be simple to implement; neither is true in our case. The most popular choice for end-to-end learning in imaging are convolutional neural networks (CNNs). There is a vast number of papers on supervised learning for inverse imaging problems; we mention a small selection [41, 40, 38, 27]. CNNs are (approximately) translation-covariant and they excel in problems that are classically solved by filtering. Examples are deblurring, denoising or inverting the Radon transform which becomes a Fourier multiplier upon a composition with its adjoint. Versatile architectures like the U-Net [39] can be applied to more general problems but they lack the right structure to capture wave physics and therefore fail out-of-dataset. A related issue with current CNN architectures is the lack of interpretability: it is not straightforward to associate different parts of a CNN with corresponding physical processes.

In the context of waves, architectures based on wavelet transforms [18] were applied to various imaging modalities [19, 20, 21]. It is however unclear whether the architecture generalizes out-of-distribution or how they compare to standard high-quality baselines such as the U-Net, which performs surprisingly well on simple generalization tasks. Finally, we point out the work on meta learning for Calderón-Zygmund operators [24]; our $\sigma$ is similar to their parameterizations.

## 2 Imaging with Fourier integral operators

The true $A_\sigma$ and its inverse can be approximated by FIOs. Our aim, however, is not to simply replicate the functionality of FIOs in a neural network. We rather follow the structure of FIOs to tease out and generalize the key components required to build a more general neural wave imaging operator. The geometry of wave propagation in a medium depends on the orientation of the elementary wave packets (see Figure 1). This suggests to decompose the input into its directional components via a bank of oriented bandpass filters. Analysis of FIOs provides a geometrically and computationally optimal choice of these filters.

### 2.1 Filtering $u$ to a box in the dyadic-parabolic tiling of Fourier space

It has been shown in [45] that for wave propagators, the so-called dyadic-parabolic tiling of the Fourier space as shown in Figure 1C is optimal [43]. Such a tiling divides the Fourier space into overlapping boxes $B_{\nu,k}$, where the length of the box is approximately square of its width. The boxes are indexed by $\nu, k$ where $\nu$ is a unit-vector denoting the orientation of each box and $k$ is its scale. We define smooth directional bandpass filters, $\hat{\chi}_{\nu,k}^2$ supported on $B_{\nu,k}$ such that they form a partition of unity, $\hat{\chi}_0^2(\xi) + \sum_{k \geq 1} \sum_\nu \hat{\chi}_{\nu,k}^2(\xi) = 1 \, \forall \, \xi$. We filter $\hat{u} := \mathcal{F}u$ into its directional components as $\hat{u}_{\nu,k}(\xi) = \hat{\chi}_{\nu,k}^2(\xi)\hat{u}(\xi)$. Note that $\hat{u}(\xi) = \sum_{\nu,k} \hat{u}_{\nu,k}(\xi)$ by definition of $\hat{\chi}_{\nu,k}^2$.

### 2.2 Geometry of FIOs: diffeomorphisms

We now show how the phase function of an FIO characterizes the geometry of wave propagation. The phase $S_\sigma$ is positive homogeneous of degree 1 in $\xi$. A Taylor expansion of $S_\sigma(y,\xi)$ in $B_{\nu,k}$ around $(y,\nu)$ is then

$$S_\sigma(y,\xi) = \left\langle \xi, \frac{\partial S_\sigma}{\partial \xi}(y,\nu) \right\rangle + S_2(y,\xi) + \text{higher order terms.} \tag{3}$$

The second-order term $S_2(y,\xi)$ varies only slowly within a box, so $\exp(\mathrm{i}S_2(y,\xi))$ can be absorbed in the amplitude $a(y,\xi)$. Following this expansion if we discretize the Fourier transform in (2) and ignore the $a$ and $S_2$ terms, then for a box-filtered $u_{\nu,k}$ we have

$$(F_\sigma u_{\nu,k})(y) \approx \frac{1}{(2\pi)^2} \sum_{\xi \in \mathbf{1}_{\nu,k}} \hat{u}_{\nu,k}(\xi) e^{i\langle \xi, \frac{\partial S_\sigma}{\partial \xi}(y,\nu)\rangle} = u_{\nu,k}\left(\frac{\partial S_\sigma}{\partial \xi}(y,\nu)\right)$$

Therefore, for each box, $B_{\nu,k}$ (or equivalently each $\nu$), the imaging operator could be coarsely approximated via a diffeomorphism or warping, $y \to T_\sigma(y,\nu) = \partial_\xi S_\sigma(y,\nu)$. We implement this warping via a bilinear resampling of $u_{\nu,k}$. Note that the medium's wave speed, $\sigma$ dictates the warping.

### 2.3 Low-rank separated representations

To get a more accurate approximation, we incorporate the amplitude $a(y,\xi)$ and the second-order term $S_2(y,\xi)$ via a low-rank separated representation [25],

$$a(y,\xi) \exp\left[\mathrm{i}S_2(y,\xi)\right] \mathbf{1}_{\nu,k}(\xi) \approx \sum_{r=1}^{R_k} \alpha_{\nu,k}^{(r)}(y)\hat{\vartheta}_{\nu,k}^{(r)}(\xi),$$

where $R_k \sim k/\log(k)$. Multiplications by $\hat{\vartheta}_{\nu,k}^{(r)}(\xi)$ act as convolutions in space, while $\alpha_{\nu,k}^{(r)}(y)$ correspond to simple (diagonal) spatial multipliers. We can now use (3) in (2) to get

$$(F_\sigma u)(y) \approx \sum_{\nu,k} \sum_{r=1}^{R_k} \alpha_{\nu,k}^{(r)}(y) \sum_{\xi \in \mathbf{1}_{\nu,k}} e^{\mathrm{i}\langle T_\sigma(y,\nu),\xi\rangle} \hat{\vartheta}_{\nu,k}^{(r)}(\xi) \hat{\chi}_{\nu',k'}^2(\xi) \, \hat{u}(\xi). \tag{4}$$

Note that the $B_{\nu,k}$'s overlap and therefore, we can generalize results of (4) to allow for the convolutions with $\vartheta_{\nu,k}$ interact across the boxes. This parallels a sum across channels in a CNN convolution layer. We denote $C_{\nu,k}(u) := u_{\nu,k}$, $A_{\nu,k}^{(r)}(w) := \alpha_{\nu,k}^{(r)} w$ and introduce $H_{\nu,k}^{(r)}(u) := \sum_{\nu',k'} \vartheta_{\nu,k;\nu',k'}^{(r)} * u_{\nu',k'}$, to generalize the convolutions with $\vartheta_{\nu,k}$. We introduce $I$

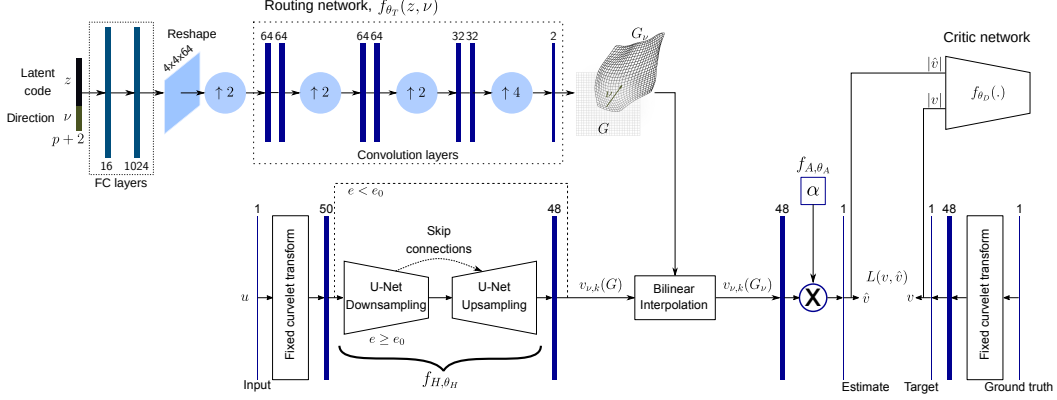

Figure 2: The FIONet. Upsampling blocks use bilinear interpolation.

as $I(w, T_\sigma(y, \nu))(y) := w(T_\sigma(y, \nu))$. Therefore, $I$ resamples $w$ via bilinear interpolation on a grid warped via $T_\sigma$. Note that $I$ is fixed and only $T_\sigma$ is learned. This gives

$$(F_\sigma u)(y) \approx \left[ \sum_{\nu,k} \sum_{r=1}^{R_{\nu,k}} A_{\nu,k}^{(r)} \circ I(H_{\nu,k}^{(r)} \circ C_{\nu,k}(u), T_\sigma(y, \nu)) \right](y). \quad (5)$$

## 3   FIONet: the architecture for wave-based imaging

The action of an FIO, $F_\sigma$ in (5) suggests incorporating spatial multipliers ($A_{\nu,k}^{(r)}$), convolutions ($H_{\nu,k}^{(r)}$), grid generators ($T_\sigma$) and directional filters ($C_{\nu,k}$) in our architecture. The dyadic-parabolic tiling of the frequency domain corresponds to the map $C_{\nu,k}$. It is a fundamental property tied to the structure of wave propagation. We thus implement it using fixed, non-trainable box filters constructed from PyCurvelab [47]. These filters correspond to the *fixed curvelet transform* layer in Figure 2 that takes an input $u$ defined on an $M \times M$ pixel grid $G$ and returns an $M \times M \times N_b$ output of spectrally filtered $u_{\nu,k}$s, where $N_b$ is the number of boxes in the tiling (see Figure 1C). We then design a convolutional module $f_{H,\theta_H}$, wave packet (WP) routing module $f_{T,\theta_T}$, and a spatial multiplier module $f_{A,\theta_A}$ with parameters $\theta_H, \theta_T, \theta_A$ such that

$$
\begin{array}{llll}
f_{H,\theta_H} & : & \mathbb{R}^{M \times M \times N_b} \to \mathbb{R}^{M \times M \times N_b R} & [f_{H,\theta_H}(w)]_{\nu,k}^{(r)} \approx H_{\nu,k}^{(r)}(w_{\nu,k}) , \\
f_{T,\theta_T} & : & \mathbb{R}^p \times \mathbb{R}^2 \to \mathbb{R}^{M \times M \times 2} & f_{T,\theta_T}(z, \nu) \approx T_{\sigma(z)}(y) \, \forall \, y \in G , \quad (6) \\
f_{A,\theta_A} & : & \mathbb{R}^{M \times M \times N_b R} \to \mathbb{R}^{M \times M \times N_b} & [f_{A,\theta_A}(w)]_{\nu,k} \approx \sum_{r=1}^{R} A_{\nu,k}^{(r)} w_{\nu,k}^{(r)} .
\end{array}
$$

The map $f_{H,\theta_H}$ operates on the entire stack of box-filtered inputs, $u_{\nu,k}$ allowing for channel interaction. Moreover, it has nonlinear activation units which generalize $H_{\nu,k}$. $A_{\nu,k}^{(r)}$ is implemented via simple multiplication layers. Here $R := R_{k_{\max}}$ is the maximum number of terms in (5) and is a hyperparameter in our training. Note that we assume that $\sigma$ belongs to a set of *natural* medium wave speeds parametrized by a low-dimensional code $z \in \mathcal{Z} \subseteq \mathbb{R}^p$ and write $\sigma$ as $\sigma(z)$. We denote the full set of trainable FIONet network parameters by $\Theta = (\theta_H, \theta_T, \theta_A)$. The network output on an $M \times M$ input $u$ is

$$\text{FIONet}_\Theta(u) = \sum_{\nu,k} f_{A,\theta_A} I(f_{H,\theta_H}^{\nu,k,:}(C_{\nu,k}(u)), f_{T,\theta_T}(z, \nu)). \quad (7)$$

### 3.1   Geometry module: warped grids and resampling

The geometry module takes as input an $M \times M \times C$ tensor and resamples each of the $C$ channels defind on the pixel grid $G$ to a new grid given by $f_{T,\theta_T}$. The WP routing network, $f_{T,\theta_T}$, is the central component of the geometry module that routes wave packets via diffeomorphisms introduced in (3).[1] Note that the grid given by $f_{T,\theta_T}$ depends on the direction $\nu$. Due to the fixed filtering map

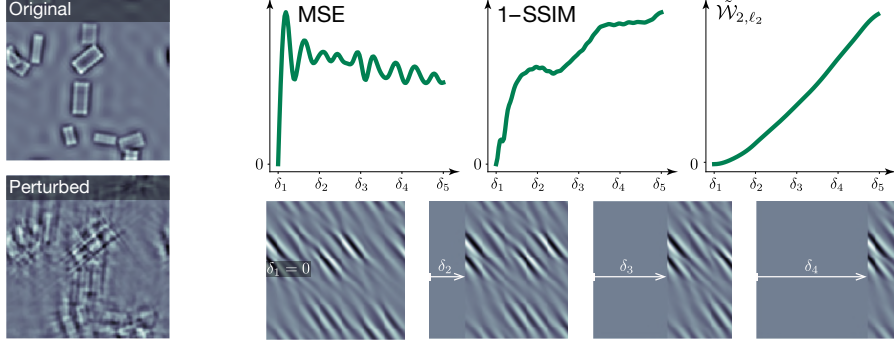

Figure 3: *Left:* Small shifts of the $(\nu, k)$ channels introduce strong distortion. *Right:* Comparison of metrics between oscillatory images: $\mathrm{MSE}(x, x_\delta)$, $\mathrm{SSIM}(|x|, |x_\delta|)$ and entropically smoothed $\tilde{\mathcal{W}}_{2,\ell_2}$ between $|x|/\|x\|_1$ and $|x_\delta|/\|x_\delta\|_1$ via Sinkhorn iteration [42].

$C_{\nu,k}$, the set of all $\nu$s is fixed. We associate each of the $C$ channels with one $\nu$; this mapping is fixed. Therefore, for each channel of the input to the geometry module, we receive a warped output grid $G_\nu := f_{T,\theta_T}(z, \nu)$ for a given $z$ (see Figure 2). The channel is then warped via the bilinear resampling operator $I$ using the grid $G_\nu$. We zero-fill if points in $G_\nu$ lie outside $G$.

An important input to the WP routing network is $z$ which represents a low-dimensional code for $\sigma$, the wave speed of the medium. The WP routing network can therefore learn the geometry for mutliple backgrounds; see Appendix E for some preliminary results. However, in this work, we assume that our all our data was generated over a fixed but still unknown wave speed corresponding to a code $z = z'$. Not knowing the wave speed leads to not knowing the exact forward operator, which therefore renders reconstruction methods like Tikhonov-regularized inverses, sparsity-promoting inverses along with more modern methods like deep image prior [50] infeasible.

Note that since the WP routing network only works on fixed $\nu$s and does not look at image pixel intensities, once trained it is data-independent. This is key for OoD generalization as the implicitly learned geometry essentially captures the effect of $\sigma$. It is absent in popular neural architectures like the U-Net [39].

## 3.2 Learning diffeomorphisms via optimal transport

We train the FIONet using a labeled set of input–output images $\{(u_i, v_i)\}$ (see (1)). We do not assume having any direct geometric information about the medium. The geometric routing information is implicit in $\{(u_i, v_i)\}$ and we aim to infer it by a suitable training strategy. We note that the warping was previously used in spatial transformers [26], but those act independently on each channel. Here it is essential that the different $G_\nu$ grids are tightly coordinated so as to get a meaningful reconstruction (see Figure 3A).

We devise a two-stage training strategy: first, we only train the geometry module which captures the bulk of the physics, i.e., we let $f_\Theta(u) = \sum_{\nu,k} I(C_{\nu,k}(u), f_{T,\theta_T}(z, \nu))$ such that an appropriate loss metric $\mathcal{L}(v, f_\Theta(u))$ is minimized. This stage is important to prevent the convolutional module $f_{H,\theta_H}$ from overfitting the training data. After $e_0$ epochs, we train the full network as in (7) using the MSE loss.

We illustrate in Figure 3 that for the first stage of training, popular metrics such as MSE for $\mathcal{L}(v, f_\Theta(u))$ fail for filtered images $u_{\nu,k} = C_{\nu,k}(u)$. Since $u_{\nu,k}$ are oscillatory $\mathcal{L}$ has many local minima which leads to the problem of cycle skipping [52, 53, 34]. SSIM is smoother but still does not give "good" gradients for training. A natural optimal transport metric based on entropically smoothed $\mathcal{W}_{2,\ell_2}$ [16] gives a consistently increasing distance metric.

While smoothed Wasserstein metrics have been used for inverse problems [2] via the iterative Sinkhorn–Knoop algorithm [11], we find that backpropagating through the iterates is unstable. We thus adopt the method of [16] and weaken the loss to an unsupervised one: instead of matching $u_i$s to $v_i$s in a paired fashion, we match the marginal distributions, $\mathbb{P}_{|v|}$ and $\mathbb{P}_{|f_\Theta(u)|}$ while, importantly, using $\mathcal{W}_{2,\ell_2}$ distance between images as the ground metric. Note that we use absolute value to ensure

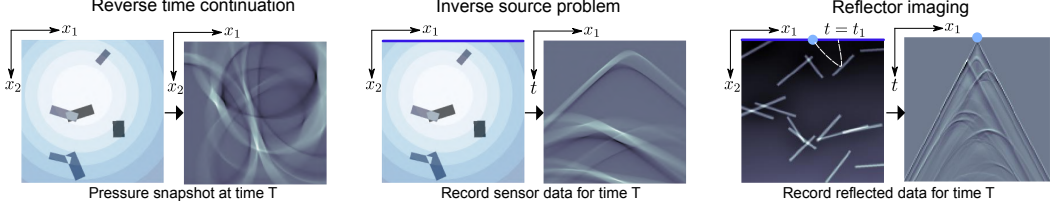

Figure 4: Three inverse problems. *Left:* Reverse time continuation: The initial pressure (boxes) propagates over the shown background wave speed. (b) Inverse source problem: Waves are recorded on the blue sensor line giving sensor traces (c) Reflector imaging: A source (blue dot) sends a pulse that is reflected at the interfaces. The dashed white line show an example ray path.

positivity of measures. As in [16], we use a critic network $f_{D,\theta_D}$ that is employed only during the stage-1 of "geometric" training. The critic network estimates the $\mathcal{W}_{1,\mathcal{W}_{2,\ell_2}}\left(\mathbb{P}_{|v|}, \mathbb{P}_{|f_{\theta_T}(u)|}\right)$, giving the final learning objective as

$$\min_{\theta_T} \max_{\theta_D} \mathbb{E}_{\hat{v}\sim\mathbb{P}_{f_{\Theta_T}(u)}} f_{D,\theta_D}(|\hat{v}|) - \mathbb{E}_{v\sim\mathbb{P}_u} f_{D,\theta_D}(|v|) + \lambda\mathbb{E}_{\tilde{v}\sim\mathbb{P}_{\text{int}}}(\|\nabla f_{D,\theta_D}(\tilde{v})\|_{\mathcal{W}_{2,\ell_2}} - 1)^2.$$

where $\mathbb{P}_{\text{int}}$ is the density generated via linear interpolations of samples from $\mathbb{P}_{|f_\Theta(u)|}$ and $\mathbb{P}_{|v|}$.

From Figure 3, we see that minor misalignments of the $(\nu, k)$ channels strongly distorts the output. Distribution matching synchronizes the diffeomorphisms to produce sharp images in the first stage of training. However, this metric only matches the distributions and not actual data pairs, hence giving us only plausible looking images. In the second stage, we train the entire FIONet (including the convolutions) using only the standard MSE loss. Much of the required geometry is already learnt in the first stage which is only tuned further via the MSE loss along with training the other modules in the network. Please see Figure 2 for the architecture details of the FIONet.

### 3.3 Modeling the low-rank separated representation by the U-Net

Finally, we implement the map $H_{\nu,k}$. We want to use standard convolutional layers. However, it is essential to ensure that we can implement the large receptive-field filters $\vartheta_{\nu,k;\nu',k'}$ of $H_{\nu,k}$. We choose to use the U-Net [39] owing to its success in convolutional tasks [27]. We give an argument on why U-Net is successful at modeling arbitrarily large filters based on the polyphase decomposition in Appendix B. Note that there are several ways to implement large filters like factorized filters, implementing filters in Fourier domain etc. In our experiments we found that U-Net was best.

**Approximating FIOs by the FIONet** While the FIONet architecture generalizes FIO, it is important to show that as a special case it can approximate exact FIOs. We make the following simplifying assumptions: 1) the WP routing network is implemented using fully connected layers; 2) the U-Net uses regular downsampling instead of max pooling. The first assumption gives us access to standard approximation theorems; in practice it only makes the forward pass slower.

**Theorem 1.** *There exists a set of weights* $\Theta = (\theta_H, \theta_T, \theta_A)$ *such that*

$$\|F[u] - \text{FIONet}_\Theta[u]\| = O(2^{-k_{min}/2})\|u\|. \tag{8}$$

This parallels [25, Theorem 2.1]. Here, the presumed sampling density in the "space" domain is naturally of order $2 \cdot 2^{k_{\max}}$ though an oversampling factor is required.

## 4 Experiments

We showcase the advantages of learning geometry and the fact that the same network architecture can be applied to various problems. We choose three inverse problems as shown in Figure 4: reverse time continuation, inverse source problem, and reflector imaging. We discuss reflector imaging and provide additional results in Appendix A. In all problems, we learn the geometry induced by the background wavespeed directly from data. In all our experiments we invoke scale separation. The coarse scale is implicit in the learned geometry. We thus aim to image the fine scales and hence high-pass our target reconstructions. The dataset and training experiment details are given in Appendix D. We choose as baseline, the U-Net, arguably the most successful architecture in imaging [40, 27].

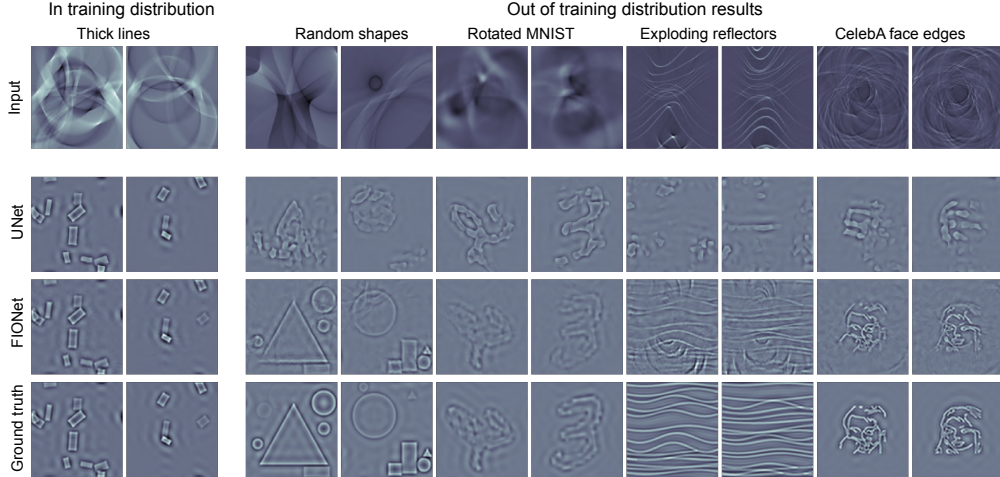

Figure 5: Reverse time continuation results. FIONet performs better in training distribution and is significantly better in out-of-distribution generalization.

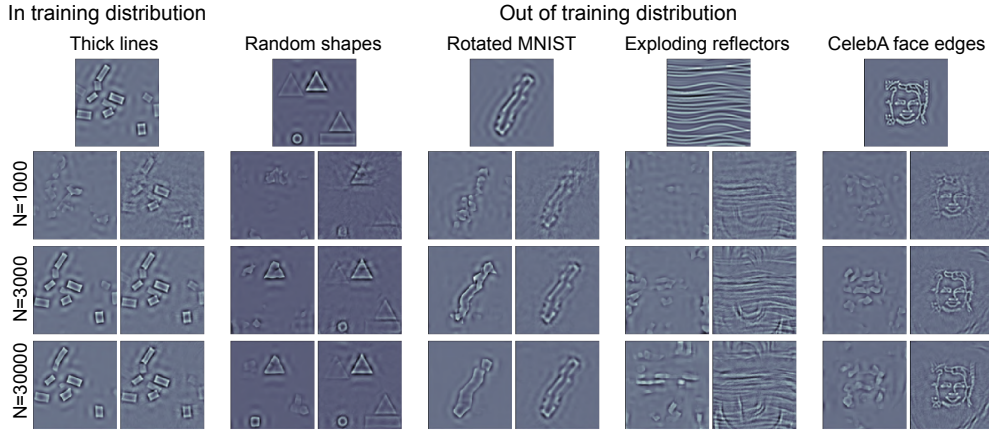

Figure 6: Inductive bias of FIONet: for each dataset the topmost row shows the ground truth, the left column per dataset shows baseline U-Net results and the right column shows FIONet results. Each row shows the result trained with $N$ samples from the training distribution.

## 4.1 Reverse time continuation

In this problem a source pressure field, $p_0$ propagates for time $T$ over an unknown background. We are given the final pressure $p_T$ at $t = T$ and we wish to estimate $p_0$. This problem most intuitively illustrates the geometry of wave propagation. Formally, it corresponds to a sum of two FIOs, one per half-wave propagation (Appendix C). We therefore train two copies of $f_{H,\theta_H}$ in parallel to model the convolutions (see (4)) in the two FIO branches, but follow them by a single WP routing network which now outputs 2 warped grids per $\nu$. In Figure 7), we show the two learned grids $G_\nu$ for each $\nu$. The outputs of $f_{H,\theta_{H,1}}$ and $f_{H,\theta_{H,2}}$ are resampled on the grids given by the WP routing network. We train on 3000 samples of randomly oriented short thick box sources and test on samples from completely different distributions. As shown in Figure 5, in the training distribution the FIONet performs slightly better than the U-Net. In out-of-distribution testing the U-Net seems to synthesize outputs from box-like patterns seen during training and therefore does considerably worse compared to FIONet. For numerical results, please see Table 1 in Appendix A.2. We attribute the out-of-distribution results to our architecture design. Due to the geometry module, the effect of the background wave speed $\sigma$ is completely captured within the learned grids $G_\nu$ which are data-independent once trained. We also

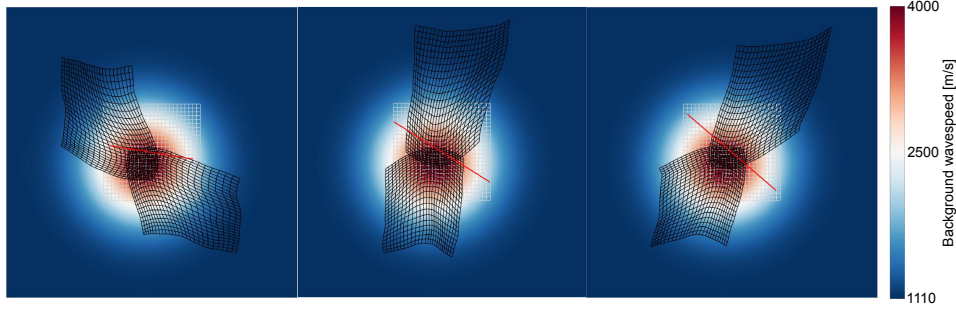

Figure 7: Diffeomorphisms learnt in reverse time continuation by the WP routing network at different orientations $\nu$ of the wave-packet - red line indicates the orientation of the wave-packet in space.

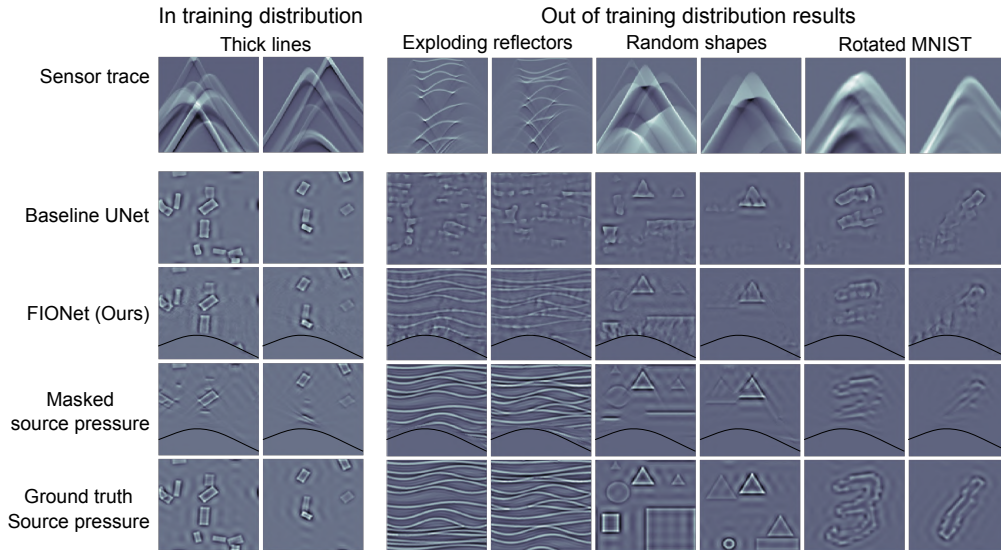

Figure 8: Inverse source problem results: We faithfully recover what the sensor sees.

study the invariance to noise in measurements in Appendix A.3 and show intermediate results from the first stage of training in Appendix A.4.

*Favorable inductive bias:* Figure 6 shows that even with a small training set (1000 samples), the FIONet achieves good performance both within and out of training distribution, which improves with the dataset size. The U-Net still synthesizes outputs using box-like patterns seen in the training set.

*Interpretability:* Since the WP routing network explicitly models the geometry of the operator, the $G_\nu$s are physically meaningful estimates (Figure 7). The deformed grids clearly show the propagation of the two half-wave solutions (Appendix C). We also found that whenever the FIONet did not give reasonable warped grids, the out-of-distribution performance suffered. This suggests that getting the geometry right is indeed central to imaging. This information is not explicitly encoded in any previous architecture.

## 4.2 Inverse source problem

In many imaging modalities (for e.g., photo-acoustic tomography, seismic imaging) sensors are placed at the domain boundary. Instead of having a snapshot of the wavefield at time $T$, we have the pressure trace at the sensor locations for all times $[0, T]$. The inverse map in such scenarios is modeled by a single FIO [29].

In Figure 8 we show how the FIONet handles such a scenario. Note that the sensor trace is in the $(x_1, t)$ domain while the source is in the $(x_1, x_2)$ domain. We deliberately choose a background such

that not all wave-packets reach the sensor boundary. In Figure 8, we show the interfaces that are "seen" by the sensor as masked source pressure. We see that these are faithfully recovered by the FIONet. Often in deep learning approaches to imaging, one claims that since the baseline U-Net reconstructs unseen data as well it is better. However, these networks can be unreliable when tested out-of-distribution [5]. Here we aim to be faithful to the physics.

The FIONet does not predict below the black line which demarcates the "seen" and "unseen" regions as dictated by the physics. Nonetheless, from the "seen" data it still reconstructs more faithfully out-of-distribution than a black-box U-Net even without knowing the background. For numerical results see Table 2 in Appendix A.2.

## 5    Conclusion and future work

We proposed a general architecture, FIONet, and a training strategy for solving inverse problems in wave-based imaging. The wave packet routing network—central to our proposal—manifests the geometry of wave propagation and scattering in its warped output grids. We showed that explicitly learning the geometry enables strong out-of-distribution generalization, outperforming competitive baselines on a variety of imaging problems. This is essential in applications of machine learning in exploratory science. FIOs model a remarkable collection of inverse problems in exploratory imaging, all of which can be addressed with the FIONet. This points to exciting opportunities in applying machine learning to relevant problems in medicine, Earth and planetary sciences, and astronomy. Our codes are available[2] for the community to reproduce our results and use our architectures for their own imaging modalities.

## Broader Impact

We do not see any major ethical consequences of this work. Our work has implications in the fields of exploratory imaging — earthquake detection, medical imaging etc. Our work improves the quality and reliability of imaging in these fields. Improving these fields has direct societal impact in finding new natural preserves, improved diagnosis in healthcare etc. A failure of our system leaves machine learning unreliable in exploratory imaging. Our method provides strong out-of-distribution generalization and hence is not biased according to the data.

## Acknowledgments and Disclosure of Funding

Authors would like to thank Herwig Wendt for the helpful discussions. MVdH gratefully acknowledges support from the Department of Energy under grant DE-SC0020345, the Simons Foundation under the MATH + X program, and the corporate members of the Geo-Mathematical Imaging Group at Rice University. ID was supported by the European Research Council Starting Grant 852821—SWING.

## Footnotes

[1]We can learn more general transformations than diffeomorphisms; for example, we can handle caustics [25].

[2] https://github.com/kkothari93/fionet

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
