[Supplementary Material]



Figure 9: Reflector imaging with a single source, in the second last row, we mask the interfaces that are not illuminated by the source for a fair comparison.

## Appendix A    Additional results

### A.1    Reflector imaging

In reflection seismology, reverse time migration is an important method for imaging the subsurface of a planet. A source pulse is sent from the surface which hits various subsurface interfaces and gets reflected back to the sensors (see Figure 13 bottom right). Again, the map from the sensor trace to the reflector interfaces is an FIO [1] given the background velocity model. We use this to learn the reflector imaging geometry and get reconstructions with a single source *without* knowing the velocity model. We highlight that in practice one uses multiple sources since not all interfaces are illuminated with a single source. Therefore in Figure 9, we also show an "illuminated-only" version of the ground-truth as well. This only shows the interfaces that were recorded by the sensors. The results are shown in Figure 9. The baseline, same as in other experiments, memorizes data specific patterns and attempts to use them to synthesize the interface distribution. We provide further quantitative results in Table 3. Interestingly, our reconstructions show migration "smile" artifacts [55] in some out-of-distribution reconstructions which is a well-known phenomenon in seismic imaging when using a single source.

### A.2    Quantitative results

We give a quantitative summary of the performance in reverse time continuation, inverse source and reflector imaging problems in Tables 1, 2 and 3 respectively. Note that since we apply entropic smoothing to the Wasserstein-2, $\tilde{\mathcal{W}}_{2,\ell_2}$ metric, we see negative values in the metric. The ordering can still be considered maintained meaning that a lower value is better. The second metric we use is the normalized cross-correlation between the images $\chi(x, \hat{x}) = \dfrac{x \cdot \hat{x}}{\|x\|\|\hat{x}\|}$. For each dataset, we choose a random sample of 20 images and report the average metric. Note that for the inverse source and reflector imaging problems, the metrics are calculated based on what data can be plausibly observed by the sensors (see second last row in Figures 8 and 9). For dataset details, refer Appendix D.

Table 1: Reverse time continuation quantitative results. All networks were trained only on the thick lines dataset.

| Dataset | Model | 1000 samples | | 3000 samples | | 30000 samples | |
|---|---|---|---|---|---|---|---|
| | | $\tilde{\mathcal{W}}_{2,\ell_2} \downarrow$ | $\chi \uparrow$ | $\tilde{\mathcal{W}}_{2,\ell_2} \downarrow$ | $\chi \uparrow$ | $\tilde{\mathcal{W}}_{2,\ell_2} \downarrow$ | $\chi \uparrow$ |
| *Thick lines* | U-Net | 33.45 | 0.08 | 1.15 | 0.92 | 3.55 | 0.81 |
| | FIONet | 18.47 | 0.72 | -7.40 | 0.97 | -4.89 | 0.94 |
| *Shapes* | U-Net | 56.84 | 0.02 | 50.80 | 0.21 | 13.71 | 0.53 |
| | FIONet | 9.67 | 0.61 | 1.71 | 0.81 | -4.74 | 0.90 |
| *Reflectors* | U-Net | 28.44 | 0.00 | 27.94 | 0.07 | 37.09 | 0.08 |
| | FIONet | 11.73 | 0.53 | 3.52 | 0.63 | 4.01 | 0.56 |
| *MNIST* | U-Net | 28.44 | 0.04 | 15.83 | 0.35 | 5.24 | 0.58 |
| | FIONet | 11.73 | 0.69 | -1.34 | 0.87 | -3.75 | 0.92 |
| *CelebA faces* | U-Net | 128.30 | 0.02 | 45.20 | 0.27 | 54.48 | 0.18 |
| | FIONet | 68.53 | 0.69 | 5.93 | 0.83 | 5.63 | 0.85 |

Table 2: Inverse source problem quantitative results. All networks were trained only on the thick lines dataset.

| Dataset | Model | $\tilde{\mathcal{W}}_{2,\ell_2} \downarrow$ | $\chi \uparrow$ |
|---|---|---|---|
| *Thick lines* | U-Net | 83.28 | 0.60 |
| | FIONet | 17.76 | 0.68 |
| *Reflectors* | U-Net | 29.06 | 0.38 |
| | FIONet | -4.02 | 0.87 |
| *Random shapes* | U-Net | 74.10 | 0.53 |
| | FIONet | 33.84 | 0.67 |
| *MNIST* | U-Net | 63.07 | 0.43 |
| | FIONet | 41.15 | 0.58 |

Table 3: Reflector imaging quantitative results. All networks were trained only on the lines dataset.

| Dataset | Model | $\tilde{\mathcal{W}}_{2,\ell_2} \downarrow$ | $\chi \uparrow$ |
|---|---|---|---|
| *Lines* | U-Net | 5.69 | 0.77 |
| | FIONet | 6.74 | 0.75 |
| *Reflectors* | U-Net | 10.22 | 0.42 |
| | FIONet | 3.17 | 0.74 |
| *Shuffled Reflectors* | U-Net | 8.50 | 0.41 |
| | FIONet | 3.09 | 0.72 |
| *Random Reflectors* | U-Net | 5.41 | 0.38 |
| | FIONet | 4.60 | 0.55 |

Figure 10: Testing 10dB noisy inputs on networks trained on clean data.

Figure 11: Reverse time continuation: output of the routing network already brings the interfaces close to where they should be.

Figure 12: The background wavespeed is perturbed by $5\%$ from the training condition and the sensor recorded. On this trace, we see that the reconstruction from the FIONet is stable in that we get the required interfaces.

## A.3 Stability under noise

We evaluate the stability of our trained networks under additive noise by testing on 10dB noisy inputs. For a comparison of performance, see Figure 5. Note that all networks were trained on clean data and then tested on 10dB noisy inputs.

## A.4 Routing network output

In order to understand how the routing network warps the image (per $(\nu, k)$ channel), we show an output from the first phase of training in Figure 11 for the reverse time continuation problem. Here the $u_{\nu,k}$s are directly warped over the grids given by the routing network. Note that here two separate half-wave solutions need to synchronize to give the final image (see Appendix C), i.e. we have two grids per $\nu$ and therefore we warp the the input channel once on each grid and sum to get the final output. We can see that the routing network already places the interfaces at almost the right locations (obviously with artifacts). The convolutional module further filters and enhances the $u_{\nu,k}$s such that after resampling on the grids given by routing network ultimately the reconstructions are close to the required image. Note that the network has never seen any image like the sample in Figure 11 during training.

## A.5 Robustness against change of background

In our experiments, the background wave speed is unkwown but fixed. In the inverse source imaging problem, we consider a case where the background wave speed at test time has about a $5\%$ deviation with respect to the training background wave speed. This is motivated by seismic applications where a $3 - 5\%$ variation in the Earth's mantle wave speed is expected [30]. We can see in Figure 12 that recovery is still stable.

## A.6 Addendum to Figure 1

Figure 13: *A:* Wave packets $S_1$, $S_2$ and $S_3$ are recorded at times $t_1$, $t_2$ and $t_3$ at the recording surface. Note that the sources travel in two directions(dashed and solid arrows)—two half-wave solutions (see Appendix C). In the inverse source problem only one half-wave gets recorded by the sensors. *B:* The sensor trace on the right shows these recordings in the $(x_1, t)$ domain. We see 3 wave-packets at $(a, t_2)$, $(b, t_1)$, $(c, t_3)$ corresponding to the arrival of $S_2$, $S_1$, and $S_3$ respectively. We also show a single sensor trace line (blue) overlayed at $x = c$. The orientation and timing of the wave packets in the trace is tied to the orientation and location of the wave packets at $t = 0$. *C:* Dyadic-parabolic decomposition of phase space. Wave packets are localized in frequency by directional bandpass filters $\hat{\chi}_{\nu,k}^2$ shown in top right. The boxes shown in green,red and blue correspond to $B_{\nu,k}$. *D:* In reflection imaging, a source emits a bandlimited pulse that is reflected at interfaces and recorded on the surface (see Appendix A.1). Notice that the ray bends *and* scatters but we only have the scattered signals.

## Appendix B    FIONet approximates FIOs: Sketch of Proof of Theorem 1

The proof of Theorem 1 can be decomposed into three parts, namely showing that

  (i) $f_{T,\theta_T}$ approximates $T_\nu$;

  (ii) $f_{A,\theta_A}$ approximates $A_{\nu,k}^{(r)}$ for all $\nu, k$ and $r$;

  (iii) $f_{H,\theta_H}$ approximates $H_{\nu,k}^{(r)}$ for all $\nu, k$ and $r$.

**Part (i)**    This follows from the results on universal approximation by deep neural networks [54] on noting that $(y, \nu) \mapsto T_\nu(y)$ is smooth in both $y$ and $\nu$. We can thus conclude that for any $\epsilon > 0$ there exists a $\theta_T(\epsilon)$ such that

$$\sup_{\nu\in[0,2\pi), y\in\mathcal{D}} \|f_{T,\theta_T(\epsilon)}(y,\nu) - T_\nu(y)\| \leq \epsilon, \tag{9}$$

where $\mathcal{D}$ is the (compact) computational domain of interest. Since we measure the reconstruction error in $L^2$, this fact alone does not immediately give us the desired estimate. We lean on results from [25] which assume that the diffeomorphisms are implemented perfectly; (9) does not yield a satisfactory bound on the $L^2$ norm since

$$\lim_{\epsilon\to 0} \|u(f_{T,\theta_T(\epsilon)}(\,\cdot\,)) - u(T_\nu(\,\cdot\,))\|_{L^2} \not\to 0$$

if $u$ is allowed to contain arbitrarily high frequencies. Conveniently, this is not true in our case: as we work with discrete pixels, we assume $u$ is adequately bandlimited before sampling. This implies that $u$ is Lipschitz continuous (pending a few technicalities: we assume $u$ is obtained as an inverse Fourier transform of a bandlimited spectrum in $L^1 \cap L^2$),

$$|u(x_1) - u(x_2)| \leq L_u \|x_1 - x_2\|,$$

where $L_u$ can be uniformly bounded by $L$ depending on the maximum norm and bandwidth. Further, we only compute the error over a compact domain. We can then write

$$
\begin{aligned}
\|u(f_{T,\theta_T(\epsilon)}(y)) - u(T_\nu(y))\|_{L^2(\mathcal{D})} &= \|u(T_\nu(y) + R(y)) - u(T_\nu(y))\|_{L^2(\mathcal{D})} \\
&\leq L\|R(y)\|_{L^2(\mathcal{D})} \\
&\leq L\sqrt{|\mathcal{D}|}\cdot\epsilon
\end{aligned}
$$

where the last quantity can indeed be made arbitrarily small since $\mathcal{D}$ is fixed.

**Part (ii)**    This follows trivially since $A_{\nu,k}^{(r)}$ is a simple linear pointwise multiplication.

**Part (iii)**    The technical difficulty in part (iii) is proving that a U-Net using small filters can approximate convolutions with $\vartheta_{\nu,k}^{(r)}$. We use a technique from signal processing called the polyphase decomposition [14, 51].

Consider a single discrete filter kernel $h[\vec{n}], \vec{n} \in \mathbb{Z}^2$ with possibly large but finite support contained within the image. We introduce the 2D $z$-transform as

$$X(\vec{z}) = \sum_{\vec{n}\in\mathbb{Z}^2} x[\vec{n}]\vec{z}^{-\vec{n}}, \tag{10}$$

with $\vec{z}^{-\vec{n}} := z_1^{-n_1} z_2^{-n_2}$. We split the image $x[\vec{n}]$ into its polyphase components $x_0[\vec{n}] = x[2\vec{n}]$, $x_1[\vec{n}] = x[\vec{l}_1 + 2\vec{n}]$, $x_2[\vec{n}] = x[\vec{l}_2 + 2\vec{n}]$, $x_3[\vec{n}] = x[\vec{l}_3 + 2\vec{n}]$, with $\vec{l}_1 = [1,0]^T$, $\vec{l}_2 = [0,1]^T$, $\vec{l}_3 = [1,1]^T$. Note that $x$ can be assembled from its polyphase components via upsampling and interleaving.

We write

$$X_l(\vec{z}) = \sum_{\vec{n}\in\mathbb{Z}^2} x_l[\vec{n}]\vec{z}^{-\vec{n}}, \quad l = 0, 1, 2, 3,$$

and find that

$$X(\vec{z}) = X(z_1, z_2) = X_0(z_1^2, z_2^2) + z_1^{-1}X_1(z_1^2, z_2^2) + z_2^{-1}X_2(z_1^2, z_2^2) + z_1^{-1}z_2^{-1}X_3(z_1^2, z_2^2).$$

and similarly
$$H(\vec{z}) = H(z_1, z_2) = H_0(z_1^2, z_2^2) + z_1^{-1} H_1(z_1^2, z_2^2) + z_2^{-1} H_2(z_1^2, z_2^2) + z_1^{-1} z_2^{-1} H_3(z_1^2, z_2^2).$$

We aim to compute $y[\vec{n}] = (x * h)[\vec{n}]$ or, in the $z$-domain, $Y(\vec{z}) = H(\vec{z})X(\vec{z})$. It is possible to write $Y(\vec{z})$ as

$$Y(\vec{z}) = \begin{bmatrix} 1 & z_1^{-1} & z_2^{-1} & z_1^{-1} z_2^{-1} \end{bmatrix} \begin{bmatrix} & & \\ & \boldsymbol{G}(\vec{z}) & \\ & & \end{bmatrix} \begin{bmatrix} X_0(\vec{z}^2) \\ X_1(\vec{z}^2) \\ X_2(\vec{z}^2) \\ X_3(\vec{z}^2) \end{bmatrix}. \tag{11}$$

such that the a priori non-unique $\boldsymbol{G}(\vec{z})$ depends only on even powers of $z_1, z_2$. That is, the corresponding filters live strictly on the subgrid $\mathcal{M} = \{ \boldsymbol{M}\vec{n} \ : \ \vec{n} \in \mathbb{Z}^2 \}$, with $\boldsymbol{M} = \mathrm{diag}(2, 2)$. When such a filter matrix is followed by regular downsampling, we can exchange the order of downsampling and filtering, by replacing $\vec{z}^2$ by $\vec{z}$.

Some deliberation shows that

$$\boldsymbol{G}(\vec{z}) = \boldsymbol{H}(\vec{z}^2) = \begin{bmatrix} H_0(\vec{z}^2) & z_1^{-2} H_1(\vec{z}^2) & z_2^{-2} H_2(\vec{z}^2) & z_1^{-2} z_2^{-2} H_3(\vec{z}^2) \\ H_1(\vec{z}^2) & H_0(\vec{z}^2) & z_2^{-2} H_3(\vec{z}^2) & z_2^{-2} H_2(\vec{z}^2) \\ H_2(\vec{z}^2) & z_1^{-2} H_3(\vec{z}^2) & H_0(\vec{z}^2) & z_1^{-2} H_1(\vec{z}^2) \\ H_3(\vec{z}^2) & H_2(\vec{z}^2) & H_1(\vec{z}^2) & H_0(\vec{z}^2) \end{bmatrix}.$$

Let

$$\vec{d}(\vec{z}) = \begin{bmatrix} 1 & z_1^{-1} & z_2^{-1} & z_1^{-1} z_2^{-1} \end{bmatrix}^T \quad \text{and} \quad \vec{x}(\vec{z}^2) = \begin{bmatrix} X_0(\vec{z}^2) \\ X_1(\vec{z}^2) \\ X_2(\vec{z}^2) \\ X_3(\vec{z}^2) \end{bmatrix},$$

and define the regular downsampling and upsampling operators as

$$(\mathsf{D}_2 \vec{x})[\vec{n}] = \vec{x}[\boldsymbol{M}\vec{n}] \quad \text{and} \quad (\mathsf{U}_2 \vec{x})[\vec{n}] = \begin{cases} \vec{x}[\frac{1}{2}\vec{n}] & \vec{n} \in 2\mathbb{Z}^2 \\ 0 & \text{otherwise.} \end{cases}$$

Noting that $\vec{x}(\vec{z}^2)$ coincides with $\vec{d}(\vec{z}^2)X(\vec{z}^2)$ on $\mathcal{M}$, we can write

$$Y(\vec{z}) = \vec{d}(\vec{z})^T \left( \mathsf{U}_2 \big( \boldsymbol{H}(\vec{z}) \mathsf{D}_2 \big( \vec{d}(\vec{z}) X(\vec{z}) \big) \big) \right),$$

with a slight abuse of the $\mathsf{D}_2, \mathsf{U}_2$ notation. This exactly corresponds to a U-Net (with identity activations and no bias) with one downsampling and one upsampling and four channels in between. The filters in the first layer are given as $\vec{d}(z)$ and they are of length at most 2; the filters in the second layer (after the downsampling) correspond to the (shifted) polyphase components of $H(\vec{z})$ so they are of length about $K/2$ for a filter $h[\vec{n}]$ with support size $K \times K$. Recursively continuing this procedure increases the number of channels by a factor of 4 and halves the filter lengths. We need about $\log_2 K$ downsampling and $\log_2 K$ upsampling layers to implement $h[\vec{n}]$ using filters of size at most $3 \times 3$. This implies that the number of channels in the innermost layer is about $4^{\log_2 K} = K^2$. We note that with ReLU activations a filtering can be standardly written as

$$h * x = \begin{bmatrix} I & -I \end{bmatrix} \begin{bmatrix} \mathrm{ReLU}(+ \, h * x) \\ \mathrm{ReLU}(- \, h * x) \end{bmatrix}, \tag{12}$$

yielding the way to insert ReLU activations in each layer. Thus, on the discretized level, the U-Net architecture can exactly represent the convolutions with $\vartheta^{(r)}_{\nu,k;\nu',k'}$.

With Parts (ii) and (iii), the FIONet reproduces (3.11) in [4] upon eliminating cross channel interaction in the filters of the U-Net. Part (i) provides an estimate of misalignment separately from this. The work of [15] provides an estimate for the approximation of (4) by (3.11) referred to above using numerical analysis which is controlled by an oversampling factor. We absorb the estimate of misalignment in this estimate. We then apply Theorem 4.1 in [12] to obtain the result using curvelets from a tight frame.

## Appendix C    FIOs and the wave equation

The Cauchy initial value problem for the scalar wave equation is given by

$$P(x, D_x, D_t)u = 0, \quad P(x, D_x, D_t) = \partial_t^2 + c(x) \left( \sum_{j=1}^{2} D_{x^j}^2 \right) c(x) \tag{13}$$

$$u|_{t=0} = h, \quad \partial_t u|_{t=0} = h', \tag{14}$$

where $D_x = -i\frac{\partial}{\partial x} \leftrightarrow \xi$ (via the Fourier transform). We summarize how to solve (13)-(14) with the plane-wave initial value,

$$h(x) \equiv 0, \quad h'(x) = \exp[i\langle\xi, x\rangle],$$

where $\xi \in \mathbb{R}^2 \setminus \{0\}$ is a parameter. To construct solutions of the initial value problem, one may invoke the so-called WKB ansatz [13],

$$u_\xi(x, t) = a_+(x, t, \xi) \exp[i\alpha_+(x, t, \xi)] + a_-(x, t, \xi) \exp[i\alpha_-(x, t, \xi)] \tag{15}$$

Invoking the initial conditions yields

$$\alpha_+(x, 0, \xi) = \alpha_-(x, 0, \xi) = \langle\xi, x\rangle \tag{16}$$

and

$$\partial_t \alpha_\pm(x, 0, \xi) = \mp c(x)|\xi|.$$

In the case that the wave speed, $c$, does not depend on $x$ we may easily find $\alpha_\pm$ and $a_\pm$ explicitly, and the WKB ansatz gives an exact solution: The eikonal equations are

$$\partial_t \alpha \pm c|\partial_x \alpha| = 0, \quad \alpha_\pm(x, 0, \xi) = \langle x, \xi\rangle$$

and have solutions

$$\alpha_\pm(x, t, \xi) = \langle x, \xi\rangle \mp tc|\xi|.$$

The transport equations are

$$\partial_t a_\pm \pm 2c \frac{\langle\xi, \partial_x a_\pm\rangle}{|\xi|} = 0, \quad a_\pm(x, 0, \xi) = \pm \frac{i}{2c|\xi|},$$

the solutions of which are simply constant

$$a_\pm(x, t, \xi) = \pm \frac{i}{2c|\xi|}.$$

Thus, in the constant wave speed case the WKB ansatz yields

$$u_\xi(x, t) = \frac{i}{2c|\xi|} \left( e^{i(\langle x, \xi\rangle - tc|\xi|)} - e^{i(\langle x, \xi\rangle + tc|\xi|)} \right) = \frac{e^{i\langle x, \xi\rangle}}{c|\xi|} \sin(tc|\xi|)$$

It is not difficult to check that when $\xi \neq 0$ this is an exact solution of the scalar wave equation with the plane wave initial data.

If we integrate $u_\xi(x, t)$ with respect to $\xi$, then we obtain oscillatory integrals in $x$ depending on the parameter $t$. The initial conditions imply that

$$u(x, 0) = (2\pi)^{-2} \int u_\xi(x, 0) \, d\xi = 0$$

and

$$\partial_t u(x, 0) = \partial_t|_{t=0} (2\pi)^{-2} \int u_\xi(x, t) \, d\xi = \delta(x).$$

Then

$$u(y, t) = (2\pi)^{-2} \int a_+(y, t, \xi) \exp[i(\alpha_+(y, t, \xi)] \frac{i}{2c|\xi|} d\xi$$

$$- (2\pi)^{-2} \int a_-(y, t, \xi) \exp[i(\alpha_-(y, t, \xi)] \frac{i}{2c|\xi|} d\xi, \tag{17}$$

yielding, at fixed time $t$, the amplitudes and phase functions of two FIOs representing the parametrices of two half wave equations. The canonical transformations, with $t$ fixed, follow from

$$
\frac{\partial \alpha_\pm}{\partial \xi} = y \mp tc\frac{\xi}{|\xi|}, \tag{18}
$$

$$
\frac{\partial \alpha_\pm}{\partial y} = y, \tag{19}
$$

yielding

$$
\left( \underbrace{y \mp tc\frac{\xi}{|\xi|}}_{=x}, \xi \right) \to (y, \overbrace{\xi}^{=\eta}),
$$

signifying straight (bi)characteristics.

## Appendix D  Training and dataset details

We generate all our data using the MATLAB kWave toolbox[48, 49]. All the scripts required to generate the data are available at https://github.com/kkothari93/fionet. We choose our computational domain to be $1024 \times 1024$ meters with a grid of size $512 \times 512$ keeping the grid spacing at 2 meters. Our background wavespeeds vary from 1400 m/s to 4000 m/s for the reverse time continuation and the inverse source problems and from 2500 to 4300 m/s for reflector imaging problem. For simplicity we chose the same background for the reverse time continuation and inverse source problems. For all the problems we choose a propagation time of $T = 200$ ms. In order to maintain CFL condition, we need a small time-step. Consequently, our sensor traces in the latter two problems are quite long - with $N > 1500$ data samples in time. To keep the computational burden under control, we subsample all our inputs and outputs to be $128 \times 128$ pixels. For the inverse source and reflector imaging problem this represents a subsampling of about 12x which affect performance. However, we find that we are still able to recover geometry.

We get the filtered directional components of our inputs using PyCurvelab [47]. We choose to have $k = 4$ scales with $1, 16, 32$ and $1$ wedges respectively per scale. For all our results, we ignore the first and the last scale completely and show results on the middle frequencies. Our method does not change even if we partition all scales. We avoid them here to keep the computational burden under control – curvelets are very redundant frames, therefore with $N$ boxes in the Fourier space one would convert the input from a single channel input to an $N$ channel input, one per box. A wedge partitioning of the Fourier space as proposed in [9] would work in our scheme in terms of learning the geometry, however, theoretically this is not the ideal partitioning for having a sparse separated representation (see Section 2).

In all the problems, unless otherwise mentioned the boundary of the domain is modeled via the standard PML (perfectly matched layer) conditions. This means that signals are not reflected back into the domain at the boundaries.

In order to illuminate a large portion of the domain in the inverse source problem, we reduce our area of interest to $[0, \frac{M}{2}] \times [\frac{M}{4}, \frac{3M}{4}]$ in a domain of size $M \times M$. Note that our sensors are at $\{0\} \times [0, M]$. Similarly for the reflector imaging problem, we reduce our area of interest to $[0, \frac{M}{4}] \times [\frac{3M}{8}, \frac{5M}{8}]$. The reflector imaging problem is more nuanced in that the rays from the reflector are unidirectional as opposed to the inverse source problem where sources propagate in an omnidirectional fashion. Therefore, seismologists would use multiple sources in order to illuminate more orientations in the image space.

The reverse time continutation problem is tested on dataset of randomly oriented thick lines, random shapes (a mix of circles, rectangles, triangles), randomly rotated MNIST digits, sinusoidal exploding reflectors (inspired from seismics) and Canny edge filtered celebA images [33]. The inverse source problem is trained on thick lines and tested on the reflectors, shapes and rotated MNIST dataset. For the reflector imaging problem, since this is mainly a seismic imaging technique we keep our datasets restricted to "layer-like" inferfaces as seen in the reflectors dataset. To simulate faults, we partition the reflectors dataset into random columns and shuffle them around. We call this shuffled reflectors dataset. Lastly we perform elastic transform on a layered medium and shuffle after partitioning into columns to get more arbitrarily shaped interfaces which we call the random reflectors dataset.

All components of our network are trained by Adam optimizer with a learning rate of $10^{-4}$. The routing network has a learning rate of $10^{-6}$. The U-Net portion of our network has 5 downsampling blocks and 5 upsampling blocks in the style of [27] with 16 starting channels that double in each donwsampling block. All activations within the U-Net are leaky ReLUs. The downsampling is done via 2D max pooling. The U-Net takes in all frequencies to allow for channel interaction as per (4). Each channel in the output of the UNet is warped as per the grids given by the routing network and then summed to give the final output. In this work, we use $R = 1$. We also find that having the multipliers do not signficantly add to the performance of the network. We run our first stage of training for 40 epochs ($e_0$ in Section 3.2) and the second stage for another 80 epochs.

The baseline network is a U-Net with everything the same except it has 6 downsampling blocks and 32 starting channels and therefore about $6 - 10$x more parameters. In our tests, these U-Net networks performed the best. The baseline is trained over 120 epochs of the training data.

All our hyperparameters were tuned only on the reverse time continuation problem based on a validation set of 100 images. The same hyperparameters were used for all three problems. For all the problems, our training set had 3000 input-output images. All results are shown on images never seen by the networks during training. This is obviously true for out-of-distribution distribution results.

**Pretraining of wave packet routing network**  A randomly initialized routing network outputs degenerate grids where almost all points are close to zero. Post resampling this leads to a almost a constant image as most pixels have been sampled from a small portion of the domain. Therefore, we pretrain our routing networks.

Obviously we do not know the background wavespeed and hence choose an arbitrary $p = 3$ parameter family of radial basis functions. The first two parameters $(z_1, z_2)$ signify the center of the Gaussian and the last parameter, $z_3$ signifies its isotropic standard deviation. We generate a training set of warped grids for different values of $z$ and pretrain our WP routing networks on those. We then fix another randomly chosen $\tilde{z}$ as an initial guess for the parametrization of background wave speed, $\sigma = \sigma(\tilde{z}$ when training for downstream imaging applications in Section 4. Note that this is done only for a good initialization of the routing networks. In fact in our experiments, we found that pretraining using constant wave speeds that do not vary over the domain also works well.

## Appendix E    Training the canonical relation from ray paths

In the process of building our architectures, we built a debugging tool which can be used to learn the geometry directly from a dataset of end points of ray-paths which is an interesting inverse problem in itself. For the reverse time continuation problem we solve both half-waves (refer Appendix C), for the inverse source problem and reflector imaging problems we solve only the half-wave from the sensor line into the interior of the domain. First, we explain the reverse time continuation problem as that is the simplest to understand due to the source and target wave-packets both being snapshots in time.

Consider any parametric family of backgrounds, $c_\theta(\cdot) : \mathbb{R}^2 \mapsto \mathbb{R}, \theta \in \mathbb{R}^p$, from which we sample $\{z_i\}_{i=1}^{N_c}$ points according to some prior $p_\theta$ (we chose uniform). For each of the sampled backgrounds, we sample $M$ phase-space points $\{(x_l, \xi_l)\}_{l=1}^M$ and simulate how wave-packets at this location in phase space would travel over the background $c_{z_i}$ by solving the Hamilton flow using the $4^{\text{th}}$ order Runge-Kutta scheme for integration. We note the final locations of these wave packets at time $T$ in phase space as $\{(y_l, \eta_l)\}_{l=1}^M$ and thus generate $MN_c$ training pairs $\{(y_k, \xi_k, z_{\lceil \frac{k}{M} \rceil}), (x_k, \eta_k)\}_{i=1}^{MN_c}$. Note that we have the *final* location and *initial* orientation as input and *initial* location and *final* orientation as output in accordance with the box algorithm laid out in [4]. We proceed by training a 4 layer fully-connected network $N_T(y, \xi, z)$ that takes in $(y, \xi)$ along with the parameter vector, $z$ and gives the estimate for $(\tilde{x}, \tilde{\eta})$. Note that due to homogeneity of the phase function, $S$ the network only cares about the direction $\hat{\xi}$ and not the magnitude. Our network layer sizes are $(4 + p, 32, 64, 128, 4)$ with leaky ReLU activations except for the last layer which has identity activation. We do not employ any normalizing or dropout strategies for training.

We also use the trained fully connected networks to generate training data for our WP routing network by calculating how an entire grid, $G$ of wave-packets oriented in direction $\nu$ would warp. This is

Figure 14: Examples of the learnt coordinate transforms trained directly from ray paths over a 3-parameter family (2 for center location, 1 for standard deviation) of simple radial basis functions. The red grid is our prediction while the black grid shows the actual deformed grid calculated using Hamilton flows. Note from the top left and bottom right figures that we can also predict caustics.

because routing network outputs entire warped grids in one pass through the network and therefore cannot be trained on ray paths directly.

We find that such a simple characterization of the canonical relation also allows for caustics to develop as shown in Figure 14 which is usually avoided in the literature on FIOs [8].

Note that the training is slightly different for the inverse source and reflector imaging problems. For these problems we need pairs $\{(y_i, (\xi_{1i}, \tau_i), z_{\lceil \frac{k}{M} \rceil}), ((x_{1i}, t_i), \eta_i)\}_{i=1}^{MN_c}$ as we pair each wave-packet seen in the sensor trace $(x_1, t)$ domain to the initial source location in $(y_1, y_2)$ domain. Here $\tau = c_z(x)|\xi| \; \forall \; (x, \xi)$ since rays follow a Hamiltonian system. In reflector imaging, the $(y, \eta)$ point corresponds to the point on the ray right after reflection.