[Reviews · NeurIPS 2020]

Review 1

Summary and Contributions: The paper studies the problem of wave-based imaging, which deals with localizing the sources of waves when they are observed after propagating through a medium of unknown properties and interfaces (routing geometry) which are central to several applications. As I understand, determining the unknown interfaces is a challenging linear inverse problem, with a measurement matrix that is known only up to a certain class, parameterized by \sigma which is an unknown value intrinsic to the medium. In order to solve this family of inverse problems, the paper advocates for a special class of NNs based on Fourier Integral Operators (FIOs) and uses optimal transport based loss function. Since the NN is motivated from a physics standpoint, the paper argues that this shows strong OOD generalization, demonstrated empirically in recovering interfaces that were not in the training distribution when evaluated on three types of inverse problems.

Strengths: * The paper presents a very interesting imaging application and how NNs can be leveraged to solve the inverse problem. As far as inverse problems themselves goes, this is more interesting than those we typically encounter in that the measurement operator itself is not fully known making it more challenging. * The proposed FIONet adds to the growing set of methods that are tightly coupled with the physics of the application of interest. There are several advantages of such a formulation — from being exactly correct (vs approximately according to some goodness measure) to being data efficient or strong OOD generalization. This paper is particularly interested in demonstrating the OOD aspect, and it is clear that using the physics is better than not (U-Net). * The proposed FIONet shows that its able to solve the different inverse problems considered.

Weaknesses: * A big concern for me is that this paper was hard to read. Since it is very applications specific, I am not familiar with a lot of the theory or the inverse problem(s) considered here. As a result, I am unable to appreciate the key aspects of the paper. For example, the introduction directly gets into the details of wave based imaging without sufficient detail or context with more commonly considered inverse problems. This makes it unapproachable for someone not familiar with this exact application. There is quite a bit of detail left in the supplement, but I believe this should be in the main paper for the contribution to be fully appreciated. * A second point about it being so applications specific is that the paper lacks context to existing methods, for e.g. how can FIONets be useful for someone outside of wave-based imaging? * Another issue is that there are no quantitative comparisons in the main paper (but in the supplement), leaving only qualitative comparisons. * There are no comparisons to any other method other than a U-Net (which essentially serves as an ablation of whether or not including the physics based network helps). Considering this is a linear inverse problem, what are other existing solutions to this problem? It is imperative to compare the proposed FIONet to iterative or classical solutions to the problem to place them in context. * Regarding the OOD experiments, this is indeed interesting because the trained network is able to give strong OOD generalization. However, particularly in imaging in the recent few years several papers have shown that untrained NNs (like deep image prior Ulyanov et al., CVPR 2018) can be used to solve inverse problems across a very wide class of images. It maybe good to mention this in the paper and place the current method in context and Ideally, also compare with those class of methods. * I am not very sure how to read or interpret figure 7 describing the diffeomorphisms. * A minor comment, there is already a model called “routing networks” (Rosenbaum et al, ICLR 2018) which are different from those described in the paper. In the interest of mitigating confusion for the reader it maybe better to clarify or re-name the model.

Correctness: Yes, they appear to be so.

Clarity: It was hard for me to follow many parts of the paper because it is very application specific (wave based imaging).. -- UPDATE -- The authors have clarified that the updated manuscript will be revised to emphasize on clarity of wave based imaging for a broader audience.

Relation to Prior Work: There is some discussion on related work, but the paper does not place itself in context to existing methods sufficiently enough to make it approachable for someone unfamiliar with the application to easily follow.

Reproducibility: No

Additional Feedback: There is a lot of discussion and details provided in the supplement, but almost nothing in the main paper. As I have mentioned in my comments earlier, a lot of reorganization is needed (esp from supplement to the main) in terms of describing the dataset or the experiments. The baselines tested here are limited, especially with respect to more traditional non-NN methods. Considering this is one of the first few papers to study this particular application, I think the onus is on it to justify why existing (non-NN) methods fail, or when they succeed in comparisons to the proposed architecture. -- UPDATE -- The author's detailed response makes some of my concerns clear, particularly regarding why other NN based methods can't be compared here due to the nature of the inverse problem. Yet, my question about comparisons with existing methods for wave-based imaging has only been cursorily addressed. Comparisons, even if they do not generalize OOD, are helpful for completeness. Even the papers referred [19,20,21] are all recent papers which are deep learning based solutions. Is it to be assumed then that there exists *no prior work* which is non-deep learning based in this wave based imaging? Having said that, I share the general optimism around the paper, and think its a very interesting use case of physics-based ML which shows strong OOD generalization properties and as such I will increase my score and recommend a weak accept.


Review 2

Summary and Contributions: The paper addresses image reconstruction for a class of inverse problems involving wave propagation. Its central focus is on building a reconstruction network that is physics-based and, therefore, explainable. The architecture of the network derives from an analysis of the Fourier integral operators (FIO), which define a large class of solutions to linear inverse problems. An oriented frequency-decomposition of the image is performed (a curvelet transform), leading to a computationally-effective approximation to FIO (based on existing multiscale techniques for inverse problem resolution). This approximation involves three operations: convolutions in the curvelet domain, image warping, and spatial weighting. The image warping operation plays a crucial role because it captures the geometry of wave propagation. It is application-specific and has a physical interpretation. The Author(s) propose a parameterization that maps a latent code and a direction to a deformation grid. They tackle the problem of learning this mapping via optimal transport. Results on numerical simulations of three inverse problems indicate a much improved generalization capability with respect to a "black-box network" (a U-Net trained to reconstruct an image from the measurements).

Strengths: The work addresses the problem of explainability of deep neural networks and the inclusion of constraints from the physics. Those are very hot topics in the NeurIPS community and this paper makes a notable contribution to the field. A notable quality of this work is the inclusion of very solid knowledge of the wave-based imaging problems to a clever learning strategy (in particular, the warping operation). A good balance is found between the presentation of the method and an analysis of the performance in several simulated inverse problems. The appendices developed in the supporting document offer an interesting physical validation of the wrapping learnt by the network.

Weaknesses: As briefly discussed at the beginning of the paper, several approaches are possible to solve an inverse problem. A mapping from the measurements to the reconstructed image can be learned: this is the approach considered both in the baseline method (a U-Net performs this mapping) and in the solution proposed by the Author(s). Classical inversion methods iteratively reconstruct the image by minimizing the discrepancy between the measurements and a model of the data that would have been collected should the image have matched the current reconstruction. These iterative methods include a forward model (that maps images to measurements). Such a model is much simpler to derive based on the physics of the problem. Deep neural networks can be involved in iterative reconstruction methods, as noted by the Author(s) in the third paragraph of page 2. The starting point for the derivation of the reconstruction network in this work is the FIO, which is a linear operator (with respect to the unknown image). It may seem quite surprising to try to mimic a linear operator to perform image reconstruction since linear methods have long been surpassed by their non-linear competitors (see for example Wiener filtering in image deconvolution vs non-linear techniques developed based on total variation minimization, sparsity in some wavelet domain, dictionary-based sparse coding, or more recently, deep-learning based deconvolution; many more examples could be given, for example in computerized tomography, MRI, etc...). Arguably, due to the many non-linearities of the neural network, the solution proposed by the Author(s) is not a linear reconstruction technique. However, the architecture of the network is derived by a close analysis and translation of the linear FIO-based reconstruction. If the aim is to learn the application-specific physics of wave-propagation, wouldn't parameterizing and learning the forward model have been a more direct path? I would really be curious to see how a simple inversion (using for example Tikhonov regularization and the forward model) would compare to the results in the paper: do the results of the proposed method match that of a standard linear reconstructor, or do the non-linearities of the network help to achieve superior performance? A possible limitation of the method is that it relies on pairs of (simulated) measurements / images. Hence, the physics learned by the network is already available through the simulator. It would be interesting to discuss how the method could apply to real data where only measurements are available (except few cases involving phantoms) and whether the physics of the imaging modality could be learned directly from the measurements. UPDATE: In their feedback, the Authors justify the reasons why these comparisons can not be fairly included in the paper. I understand their point of view and would like to tone down my criticisms. The theoretical and numerical study of their method seem solid to me and would justify the paper being accepted. I recommend that the Authors include in the main paper a more extended discussion about the problem of estimating the component "sigma", related to the geometry of the problem: how this problem is handled in the literature, how it is set in the training phase and how well their network generalizes with respect to this component (to complement the discussion with respect to the reconstruction of the images of interest and the generalization capability with respect to another distribution of images). There are elements in the supplementary document that I think should be summarized and pointed to in the main paper so that the Readers better grasp this important aspect of the problem.

Correctness: The mathematical development and physical modeling seem sound to me. I must say however that I was quite surprised that the Author(s) used a U-Net in order to model a (multi-channel) convolution. I did not find the reason indicated by the Author(s) (the need to learn large kernels) to be convincing. There are many ways to learn larger kernels (separable filters, dilated convolutions, recursive filtering, factorization as a cascade of small filters, ...) and using a non-linear network to learn such a simple linear operation seem inadequate if the aim is to approximate at best the convolutions. Of course, using a non-linear network makes the reconstruction non-linear which has advantages in itself (cf. my comment on the linearity of FIO) and could justify the choice made. I would really appreciate if a linear reconstruction method could be included as a baseline method to strengthen the claims in the experimental section. UPDATE: Based on the discussion in their feedback, I would recommend that the Authors put less emphasis on mimicking the FIO and discuss a little bit more the advantages of using a non-linear block such as a U-Net, as supported by their numerical experiments.

Clarity: I found it hard to follow at some places the mathematical development due to insufficient details on the notations. The dimensions and meaning of the operators in eq. 5 are missing (e.g., C is a cropping operation, H a sum of convolutions, A a weighting). The meaning of M is given in section 3.1 while it is first used in the previous paragraph. The equation that defines FIONet right after equation (6) is hard to follow, I would suggest explicitly adding an interpolation operator to make it clear that the image output by f_H is sampled at the location returned by f_T and multiplied by the amplitude f_A at location y. The dimensions of the three operators f do not match in the equation. Given the central role of this equation (it defines the reconstruction operator of the proposed method), no room should be left for ambiguity. The section 3.2 describes the use of Wasserstein metrics. The notations W_{2,\ell_2} are not defined (which can be understood, given the space constraints). Please indicate a reference or better still, give more details in the appendices about this notation and the underlying theory to readers that are not familiar with the topic. Appendix E is a very interesting addition to the paper and would deserve more emphasis in the main paper. UPDATE: The Authors plan several modifications / corrections to the paper that should address the issues I raised.

Relation to Prior Work: I found the relation to prior work to be satisfying.

Reproducibility: Yes

Additional Feedback:


Review 3

Summary and Contributions: In this paper, the authors study wave-based imaging problem, and propose a deep learning architecture based on discretization of Fourier Integral Operators (FIO). Towards that, they derive a mathematical expression to approximate the inverse imaging problem using FIOs and design a convolutional neural network (CNN) architecture to learn this inverse mapping from training data. The key module in this architecture is the so-called "routing network" where wave-packets are mapped back where they were created/scattered. Moreover, an optimal transport based loss function is employed to train the network. The authors illustrate the results on different inverse problems (reverse time continuation, inverse source problem, reflector imaging) and on both training and out-of-training data distributions. Almost in all experiments, FIONet outperforms -the prior work used for comparison in the paper-.

Strengths: I would list the strength of this paper as follows. - The paper in general is clearly written, the mathematical expressions are derivations are easy to follow. - The methodology proposed for inverse imaging problem outperforms the studied prior works on various datasets both in training and out-of-training distributions.

Weaknesses: My main concerns with this paper are as follows. - The training data used in the experiments are clean (noiseless). I believe that is very limiting, especially in imaging problems, where noise is actually present. Moreover, strong generalization claim sounds a bit too strong when the network is trained with clean images. For example, in interferometer imaging, to which I believe this paper applies to, artifacts often mislead the imaging methods as if they are actually source. Especially for cases where source has elliptic characteristics. Therefore, I believe the experiments are limiting in this regard. What authors think about using the clean images while training? I am eager to correct myself if there is a natural reason why clean images are used in training. - The main contribution of this paper, in my opinion, is the modeling of wave-based imaging problem that is based on discretization of Fourier Integral Operators (Equation 5). Although CNNs are used, I believe this paper intellectually contributes to image processing field more than machine learning. *UPDATE* In their response, the authors pointed out the noisy experimental results in the Appendix, which I find quite convincing in terms of capability of their approach in the noisy setting.

Correctness: I believe the correctness of the method. Regarding empirical methodology, as I have pointed out earlier, I am doubtful about the use of clean images for training. If I were the authors, however, I would have specified my claim on "strong out-of-distribution generalization" a bit more clearly. That is, there is no theoretical base for this claim, but numerical observations that suggest this. *UPDATE* In their response, the authors agreed to emphasize that their findings that suggest strong generalization is based on the empirical observation.

Clarity: I think the paper is overall well written. I have a few comments: - I recommend putting some crucial experimental details (on training and dataset) to the main body instead of the supplementary material. - I believe the authors could have used less number of specific terms as it sometimes cuts the flow for reader.

Relation to Prior Work: It is clearly discussed (together with the supplementary material) how this work differs from certain prior work. However, I am not clear what is the reason behind the lack of comparison to other imaging methods that are not based on neural networks. For example, could comparison be made to compressive sensing methods? Non-specialized CNN? It would be helpful if the authors can comment on that. - "It is however not clear whether the various components indeed generalize out-of-distribution or how they compare to standard high-quality baselines such as the U-Net, which performs surprisingly well on simple generalization tasks." Could authors explain why the comparison to U-net is unclear? *UPDATE* In their response, the authors clarify why the method is not compared to certain other methods, namely, due to the lack of knowledge on the transform operator.

Reproducibility: Yes

Additional Feedback: After the rebuttal period, I might change my assessment based on the response of authors. *UPDATE* Based on the authors response, I am clarified about my concerns and hence recommend acceptance of the paper.


Review 4

Summary and Contributions: The manuscript introduces a deep neural network architecture, FIOnet, for approximating Fourier integral operators (FIOs) that arise in various wave-based imaging problems. This network differs from prior approaches by including an routing network, a component that explicitly characterizes the spatial warping induced by the background medium on the wave packets. The authors show how such a network may be trained by first minimizing a Wasserstein loss with respect to the warped images, followed by a conventional end-to-end training of the full network. In addition, the paper shows that the proposed network is general enough to approximate a given FIO. Finally, the performance of the network is evaluated over a set of wave-based inverse problems with satisfactory results compared to a U-Net baseline.

Strengths: The direct characterization of the geometry of the imaging problem in the form of a routing network is very interesting and could provide a powerful tool for other types of problems. Performance on the tasks selected for evaluation is also very impressive. The construction of the FIOnet is explicitly derived from the general structure of the FIO itself through a series of approximations, ensuring that the network, in particular the routing network component, is able to take advantage of the underlying physics of the problem. This approach also seems to increase robustness to out-of-distribution samples.

Weaknesses: The motivation and description of the method is very high-level, presuming a great deal of background knowledge in wave-based imaging. For example, many approximations made in constructing the FIOnet architecture (such as those in eq. (3) and related to eq. (4)) are rather superficially motivated and hard to follow. The description of the network is mostly relegated to Figure 2, which comes without much explanation. The training process is similarly hard to follow in its use of Wasserstein losses and critic networks. As a result, the paper is less relevant to the NeurIPS community, even though the ideas and results presented are interesting more broadly. If the authors had presented their method in more detail (preferably in the manuscript, but if necessary, in the appendix), that would have significantly alleviated the issue. UPDATE: The authors have stated that they will improve the clarity of the paper by reorganizing it and reworking the problematic sections.

Correctness: As mentioned above, there are certain details missing in the derivation of the method, but on the whole, it looks to be well-motivated. It also seems to perform well on the simulated data provided for the different imaging tasks. In the empirical evaluation, however, the authors only compare their results to the (rather generic, albeit powerful) U-Net architecture. Several other works, in particular those of Fan and Ying, are mentioned in the introduction, but there is no comparison to any of these methods. There are also no comparisons with traditional imaging techniques that do not rely on learning. UPDATE: As pointed out by the authors in the rebuttal, it is hard to compare with other methods, since they assume a known forward operator. That being said, it might still be interesting to show this comparison as a sort of oracle result, while making it clear that the proposed method cannot be expected to outperform it due to the more limited knowledge available.

Clarity: The paper is well organized, although several details are unclear. For one, there is the lack of detail described above, but other issues regarding notation and definition of various quantities also pose problems. As an example, eq. (4) replaces an integral with a (presumably discrete) sum over ξ. That same equation also refers to the function Θ_ν,k;ν',k', which has not been previously defined. Similarly, in Section 3, the definition of the FIOnet relies on many variables, M, N_b, R, p, and z, which only defined much later (if at all). This makes the paper quite hard to follow. UPDATE: The authors have agreed to reorganize and rework parts of the text to improve clarity and rigor.

Relation to Prior Work: Previous work is mentioned in the introduction, but there is not much comparison beyond that with the baseline U-Net. Again, it would be useful to compare architectures and results with other architectures specifically targeted to wave-based imaging.

Reproducibility: No

Additional Feedback: The figures in the paper take up quite a bit of space. I would suggest the authors compress them before submission.

[Author Response · NeurIPS 2020]

We thank the reviewers for their detailed comments. We first address a few common queries and then follow with comments specific to individual reviewers.

**Linearity and comparisons**: Several reviewers comment on a lack of comparison with classical methods. We emphasize that the background wave speed, $\sigma$, and, hence, the forward operator $A_\sigma$ are *unknown* (lines 37-38). The dependence of $A_\sigma$ on $\sigma$ is very nonlinear. As pointed out by Reviewer 1, this makes the problem a challenging and novel variant of traditional linear inverse problems, out of reach of the classical approaches. (In practice $\sigma$ in, say, reflector imaging is inferred separately using, crucially, multiple sources rather than just one as in our paper.) It is truly a problem that was unlocked by machine learning. Unfortunately, most iterative or learning-based solutions assume a known forward operator. This is in particular true for the deep image prior and Tikhonov- and sparsity-regularized inverses. There are thus no fair comparisons that can be made with such methods. Further, the true forward operator in our inverse problems is non-linear; our networks are modeled after a certain linear approximation but are themselves strict non-linear generalizations. The closest papers we refer to, for example, Fan and Ying, assume a smooth *known* background. We compare to the U-Net because the U-Net is incredibly successful on a variety of imaging tasks including the ones we address, regularly outperforming the classical methods. Our main message is that this (if we may say so) *annoying effectiveness* is limited to the training dataset for hard, very non-convolutional wave imaging problems. By modeling the physics via the routing network, the FIONet robustly generalizes out-of-distribution (OOD).

**Clarity/reorganizing**: Several reviewers state that the paper would benefit from reorganization. Our first strategy was to describe the core novelty and FIO in the main paper and give technical details in appendices. The reviews make it clear that this is unfriendly to people unfamiliar with wave imaging. Thankfully they show a clear path towards improving clarity: move some appendix material to main text (explanations and quant. results), expand the intro on wave imaging, and move some figures and derivations to the appendix. These changes are not hard and we would be pleased to effect them. We thank Reviewer 2 for pointing out issues with notation which we addressed, and to Reviewer 4 for specifying the hard-to-follow parts. We are confident that the reorganization will make things clearer.

**Reviewer 1—FIONet outside wave imaging?** We reiterate that wave-based imaging encompasses a vast range of modalities in practice today (lines 25-27, 42-44). Therefore, we do not find the focus on wave imaging particularly limiting. That said, the routing network idea applies whenever there is direction-dependent transport, for example in fluid flows or ray tracing. **Figure 7**: Figure 7 shows how wave packets oriented along different directions, $\nu$ (red line in the Figure) move over a given background. The two warped grids show two directions of propagation after time $T$ (Appendix C explains the two directions). The learned warping is exactly what the physics dictates. **Renaming**: Thank you for pointing us to the original "routing network". We will add a comment and use "wave packet routing networks".

**Reviewer 2—Large filters and the U-Net?** The main reason for the discussion of how the U-Net implements large filters is theoretical rather than practical: we argue that the FIONet approximates FIOs by connecting to earlier analytic results. You are right (and we will emphasize this better) that there are many ways to implement large filters. The practical reason we chose the U-Net is that, empirically, it excels at convolutional and to some extent pseudodifferential problems. One may say that the routing network unwarps the input in such a way that it is a simple job for the U-Net. We did try a variety of other architectures including dilated convolutions and direct implementation of filters in the Fourier domain; the U-Net performed the best. **No paired data?** Training with $y$s alone might be accomplished via self-supervision. Usually $A$ is known, but knowing it up to a class could suffice! This is exciting, thank you! We will try it out asap, but it is out of the scope of this paper which introduces the interpretable geometric architecture. Let us add that strong OOD generalization allows us to train on simulated data and test on real data. We can thus learn to solve the problem "offline" and then apply to real measurements without fear of overfitting.

**Reviewer 3—Noisy Measurements**: We do have results for noisy measurements in Appendix A.3. We trained on clean data and tested on 10dB noisy data. Note that in our problems the inverse is $L^2$-stable so additive noise will not cause big problems. Training with noise improves results but not dramatically; we would be happy to add those results. The U-Net is excellent at removing additive noise and artifacts but it cannot handle geometry even without noise. **Artifacts in interferometer imaging:** Indeed FIOs associated with canonical graphs can exhibit artifacts in case of caustics (Appendix E). In fact, a single source in our reflector imaging experiment develops caustics. Interestingly, we show in Appendix E that the FIONet handles it well. The explanation is that our routing network can learn arbitrary warpings, not just diffeos. (The training data was generated with kWave which simulates full wave physics, so we commit no inverse crime.) **Empirical OOD:** Though motivated by theory our OOD findings are indeed empirical. We will emphasize this. **Relevance to ML**: There is a burgeoning interest in physics-inspired deep learning architectures as pointed out by Reviewer #2, as well as in OOD generalization. **Comparison to Fan and Ying**: While refs. [18,19,20,21] are also physics-inspired, they address pseudodifferential operators (contained in FIOs) with simple geometry (singularities do not propagate). Their OOD experiments are combinations of training scenarios (e.g., 2 triangles in training, 4 in testing), whereas we completely change the data. They do not compare to a strong learning baseline like the U-Net which would almost certainly generalize well.

[Meta-Review · NeurIPS 2020]

All four reviewers participated in the discussion and found the rebuttal somewhat satisfying. Some of the reviewers increased their score. The reviewers agreed that the paper should be accepted as the FIOnet architecture represents a solid contribution. There were also some criticisms about the paper, specially regarding comparison with other methods. I would like to encourage the authors to follow the reviewers' suggestions to improve the camera-ready version. In particular, I strongly encourage the authors to include a comparison against an oracle method as suggested by R4.